# Chalcogen Noncovalent Interactions between Diazines and Sulfur Oxides in Supramolecular Circular Chains

**DOI:** 10.3390/ijms25137497

**Published:** 2024-07-08

**Authors:** Emna Rahali, Zahra Noori, Youssef Arfaoui, Jordi Poater

**Affiliations:** 1Laboratory of Characterizations, Applications and Modeling of Materials (LR18ES08), Department of Chemistry, University of Tunis El Manar, Tunis 1068, Tunisia; emna.rahali@etudiant-fst.utm.tn (E.R.); youssef.arfaoui@fst.utm.tn (Y.A.); 2Department de Química Inorgànica i Orgànica & IQTCUB, Universitat de Barcelona, Martí i Franquès 1-11, 08028 Barcelona, Spain; zahranoori2009@gmail.com; 3ICREA, Passeig Lluís Companys 23, 08010 Barcelona, Spain

**Keywords:** chalcogen bond, density functional theory (DFT), energy decomposition analysis, noncovalent interaction

## Abstract

The noncovalent chalcogen interaction between SO_2_/SO_3_ and diazines was studied through a dispersion-corrected DFT Kohn–Sham molecular orbital together with quantitative energy decomposition analyses. For this, supramolecular circular chains of up to 12 molecules were built with the aim of checking the capability of diazine molecules to detect SO_2_/SO_3_ compounds within the atmosphere. Trends in the interaction energies with the increasing number of molecules are mainly determined by the Pauli steric repulsion involved in these σ-hole/π-hole interactions. But more importantly, despite the assumed electrostatic nature of the involved interactions, the covalent component also plays a determinant role in its strength in the involved chalcogen bonds. Noticeably, π-hole interactions are supported by the charge transfer from diazines to SO_2_/SO_3_ molecules. Interaction energies in these supramolecular complexes are not only determined by the S···N bond lengths but attractive electrostatic and orbital interactions also determine the trends. These results should allow us to establish the fundamental characteristics of chalcogen bonding based on its strength and nature, which is of relevance for the capture of sulfur oxides.

## 1. Introduction

Noncovalent interactions play a crucial role in various scientific fields such as chemistry, biology, and material science [1,2]. While hydrogen bonds are well-known [3], there are other significant noncovalent interactions including halogen [4,5,6,7,8], chalcogen [9,10], pnictogen [11,12], and tetrel bonds [13,14]. These interactions are highly directional and form between the electron-deficient region of a covalently bonded atom from groups IV-VII of the periodic table and a negative site, such as an anion or Lewis base. These interactions are collectively known as σ-hole interactions because of the uneven distribution of atomic charge on the participating atom, resulting in an electron-deficient region termed the σ-hole [15,16]. The nature of σ-hole interactions has been assigned to be primarily electrostatic, and they involve donor-acceptor orbital interactions [17,18]. Noticeably, very recently, de Azevedo Santos et al. proved that HOMO-LUMO orbital interactions, i.e., covalence, provide a determinant contribution to the bond energies of these bonds, besides the electrostatic attraction [19]. The significance of σ-hole bonding is evident in various applications such as molecular recognition [20,21], biological activity [22,23], crystal engineering [24,25], and the development of transportation and sensing technologies [26,27]. These interactions contribute to the understanding and design of complex molecular systems and materials, highlighting their importance in advancing scientific and technological progress [28].

While the σ-hole refers to the positive region of a molecule along the extension of the covalent σ-bond, there is also evidence of a positive region perpendicular to the σ-framework of the molecule, known as the π-hole [29]. The π-hole is another type of directional noncovalent interaction [18]. A notable example of the π-hole interaction is observed in sulfur dioxide (SO_2_), where the depletion of charge density in unoccupied p-type orbitals on the central sulfur atom results in a positive molecular electrostatic potential (MEP) region perpendicular to the molecular plane [30,31,32]. This phenomenon has been confirmed through MEP isosurfaces, which illustrate the presence of positively charged regions above and below the plane of the molecule. A notable work in this regard is that by Murray and Politzer, which reviewed the relevance of MEP in elucidating the nature of both σ-hole and π-hole bonding [32,33,34,35].

Numerous studies have explored π-hole interactions and investigated ways to modulate their strength [36,37,38,39,40,41]. These interactions, like σ-hole interactions, have significant implications in various fields, contributing to our understanding of molecular behavior and aiding in the design of advanced materials and biological systems.

The occurrence of π-hole-based tetrel and chalcogen bonds has been observed in many experimental studies [42,43]. Notably, the chalcogen bond is a versatile interaction capable of participating in complex formation through either π-hole or σ-hole interactions. Chalcogen bonds are distinguished by their strength, which can be comparable to or even exceed that of hydrogen bonds. For instance, chalcogen-bonded complexes formed between the sulfur atom in FHS and the nitrogen atom in NH_3_ result in a robust S···N noncovalent bond with a binding energy of approximately 8 kcal/mol [44]. In a study by Mehdi D. Esrafili et al. [39], the formation of S···N chalcogen bonds through π-holes was examined in SO_3_ complexes with various nitrogen-containing molecules such as NH_3_, H_2_C=NH, NH_2_F, NP, NCH, NCF, NF_3_, and N_2_. Additionally, S. Bhattarai investigated π-hole chalcogen-bonded complexes between substituted pyridines and the SO_3_ molecule, identifying two types of chalcogen-bonded complexes [45]. Type-A complexes are formed via lone pair π-hole (N···S) interactions, while Type-B complexes are stabilized through π-hole–π-electron interactions, highlighting the significant role of π-hole chalcogen bonds in molecular interactions and complex formation.

Sulfur oxides are trace atmospheric components that play significant roles in the earth’s geology, biology, and climate, making the study of their interactions with other compounds crucial [45,46,47,48]. Investigating sulfur oxide-diazine complexes is therefore highly relevant for understanding the behavior and stability of noncovalent complexes and their potential applications across various fields. This research not only advances our comprehension of noncovalent interactions but also contributes to the development of theoretical models and explores the biological and practical implications of sulfur oxide interactions.

Expanding our systems to chains consisting of six to twelve monomers is crucial for exploring the impact of these larger molecular structures on intermolecular interactions. This investigation seeks to elucidate the alterations in both the strength and nature of noncovalent interactions, chalcogen bonds, in particular, among the monomers within these extended chains. Numerous studies have extensively explored the interaction between sulfur oxides and pyridine molecules, with a specific focus on the N···S interaction, to assess the potential applications of these interactions in the chemical and biological fields [45,46,47,48]. However, there has been limited research on the interaction between sulfur oxides and diazines. This research gap prompted us to investigate the chalcogen bonding between sulfur oxides and diazines.

Thus, in this present study, we will mainly focus on N···S chalcogen bonds involving π-holes between diazine molecules (pyrimidine, pyridazine, and pyrazine) as Lewis bases and sulfur oxides (SO_2_ and SO_3_, Figure 1) as Lewis acids [49]. Understanding these interactions should contribute to the design of supramolecular sensors aimed at capturing and eliminating sulfur oxides through chalcogen bonding interactions. For this, our three main objectives are the following: (1) characterize the N···S chalcogen bonds, formed via π-holes between diazine molecules and sulfur oxides, by investigating their nature and strength; (2) analyze the interaction energies with the aim of comparing the stability and strength of chalcogen bonds in different diazine–sulfur oxide complexes; and (3) analyze the MEP isosurfaces to identify the positive regions (π-holes) on sulfur atoms in SO_2_ and SO_3_ and their interaction sites on diazine molecules. This study is based on a dispersion-corrected DFT Kohn–Sham molecular orbital analysis together with quantitative energy decomposition analysis.

## 2. Results and Discussion

### 2.1. Interaction between Diazines and SO_2_

We first discuss the supramolecular assemblies between the three diazines and SO_2_. In the case of pyridazine, all circular molecular assemblies (n)pyridazine/SO_2_ with n = 2–6 adopt a quasi-planar conformation with C_nv_ symmetry (Figure 2). Importantly, sulfur atoms are in the same plane as the pyridazine rings, with a distance that increases from 2.718 to 2.931 Å from n = 3 to 6, respectively. The longer distance in the case of 2pyridazine/SO_2_ than for either n = 3 or 4 is due to its more strained structure. For the same reason, in the case of either pyrimidine or pyrazine, the species with n = 2 is not stabilized because of the too-close proximity and thus repulsion between the two diazine rings. The structures formed with pyrimidine also adopt C_nv_ symmetry, but the rings are more bent, escaping from the quasi-planar conformation (Figure 3). Opposite to pyridazine, the S···N bond length decreases from 2.829 to 2.750 Å from n = 3 to 6, respectively. The same trend is also given by the third group with pyrazine, with the S···N bond length decreasing from 2.814 to 2.720 Å from n = 3 to 6, respectively (Figure 4). The structures formed with pyrazine present D_nh_ symmetry, with a complete perpendicular conformation of the rings with respect to the formed circular geometry with SO_2_.

Having discussed the geometries, we now focus on the bonding energies of this series of supramolecular species. The bonding energies increase with the size of the species (Table 1). For each value of n, ∆*E* decrease from pyridazine to pyrimidine to pyrazine. For instance, in the case of n = 3, the value decreases from −46.1 to −37.4 to −33.4 kcal mol^−1^, respectively. Based on these values, we could assign the stronger interaction for 3pyridazine/SO_2_ to its shorter distance. However, in the case of n = 5, ∆*E* also decreases from −72.0 to −71.5 to −63.8 kcal mol^−1^ from pyridazine to pyrimidine to pyrazine, respectively, despite the S···N bond length of 5pyridazine/SO_2_ being longer. System 6pyridazine/SO_2_ is the only exception. In addition, the trends are not determined by the strain energy of the fragments (∆*E*_strain_), which also increases with larger n, but with a maximum value of 3.9 kcal mol^−1^ (Table 1). Thus, the distance between SO_2_ and the diazine rings is not the only factor responsible for the computed bonding energies, and further insight into the interaction energies is needed.

But first, to make an easier comparison, the trends in the bonding energies with n can be better seen if the ∆*E* values are divided by the number of interacting species in each case (2n, i.e., 6 for n = 3, 8 for n = 4, 10 for n = 5 and 12 for n = 6). Also in this case, the bonding energy per unit decreases from pyridazine to pyrimidine to pyrazine (Table 2). However, whereas those for pyridazine decrease from n = 3 to 6, the opposite is observed for both pyrimidine and pyrazine. These latter trends agree with the shortening of the S···N bond length in the case of pyridazine and the lengthening in the case of the two other diazines. However, this S···N bond length is not responsible for the difference between the different diazines, as stated above.

With the aim of understanding the interaction in this series of systems, we performed a Kohn–Sham molecular orbital analysis together with a quantitative energy decomposition analysis (EDA) [50,51,52,53,54]. First, the interaction under analysis can be described as a covalent interaction, as the orbital interactions are more than half the magnitude of the electrostatic interactions [19]. In particular, if we focus on the attractive interactions (∆*V*_elstat_ + ∆*E*_oi_ + ∆*E*_disp_), the electrostatic interactions are in the order of 49–52%, whereas the orbital interactions are in the order of 30–34%. Next, in general, larger interaction energies go with larger repulsive Pauli and attractive electrostatic and orbital interactions that compensate for the latter and make the interaction attractive (Table 3). This is perfectly observed for n = 3 and 4 with pyridazine, which presents larger ∆*E*_Pauli_ than pyrimidine or pyrazine. However, from n = 5, the shorter S···N bond lengths of these two diazines cause an increase in repulsive ∆*E*_Pauli_ that, together with more attractive ∆*V*_elstat_ and ∆*E*_oi_ terms, makes their interaction become stronger than with pyridazine. It can also be observed that the differences in the EDA terms between pyrimidine and pyrazine are small, and for this, further analysis is needed.

We can gain insight into the EDA analysis by focusing on the interaction between one molecule of diazine and SO_2_ and on the same geometry as the whole circular system [55,56]. The corresponding EDA values of these diazine···SO_2_ show the effect of either elongating the S···N bond length in the case of pyridazine or shortening it in the case of both pyrimidine and pyrazine (Table 4). The systems with pyridazine show a stronger interaction in all cases except for 6pyrimidine. The main determinant term is the Pauli repulsion, followed by the electrostatic interaction and then the orbital interaction. Dispersion interaction remains constant for all systems. Again, with respect to the comparison of the interaction between pyrimidine and pyrazine, small differences in favor of the former arise from the combination of slightly more attractive electrostatic and orbital interactions. For completeness, the above discussion can be complemented by the EDA analysis with the same methodology, i.e., analyzing the terms divided by the number of molecules for each interacting system (Appendix A), showing the same trends.

Having discussed the role of the distance between the diazine rings and SO_2_, i.e., the S···N bond length, and its effect on the interactions, we now focus on the electrostatic interaction that has also been observed to play a determinant role in the trends. For this, we computed the VDD charges and depicted the molecular electrostatic potential (MEP) isosurfaces of the involved systems (Figure 5). The chalcogen bond formed between diazines and SO_2_ can be attributed to the interaction between the positive MEP region (i.e., the π-hole) on sulfur, depicted in blue, and the negative MEP region of the nitrogens, depicted in red, in diazines (Figure 5). Pyrimidine presents the most negatively charged nitrogen atoms (−0.193 au) among the diazines, which cause a more favorable interaction with the positively charged S atom of SO_2_ (+0.478). This is the reason why pyrimidine systems present a more attractive interaction with SO_2_ than those with pyrazine, as just discussed above. In addition, from the MEP isosurfaces of the whole systems (Appendix A), we can observe the charge transfer from the diazines to the SO_2_ molecules, specifically from the lone pair of the nitrogen to the π-hole of SO_2_. Consequently, this results in a decrease in the negative charge on the nitrogen atoms and a simultaneous increase in the charge on SO_2_. For instance, in the case of 3pyridazine, the N charge is decreased from −0.115 to −0.058 e, for 3pyrimidine, from −0.193 to −0.157 e, and for 3pyrazine, from −0.166 to −0.134 e (Table 5). This charge transfer is reduced with larger systems (the N charge is −0.084, −0.143, and −0.114 e for 6pyridazine, 6pyrimidine, and 6pyrazine, respectively). In all cases, pyrimidine gives the most attractive electrostatic interaction. However, a direct comparison among the systems must also consider the S···N bond length, as stated above.

The other attractive interaction that plays a role is the orbital interaction, as discussed above. The main interaction is found between the HOMO of the diazine ring and the LUMO of the SO_2_ molecule (Figure 6). The former mainly involves the sigma lone pairs of the N atoms of the rings (pyr^HOMO^), whereas the latter mainly involves the π lone pairs of S and O atoms (SO_2_^LUMO^), both being antibonding. Thus, these supramolecular circular systems under analysis are mainly determined by a donor–acceptor interaction between the diazine ring and the SO_2_ molecule. The overlaps between these two orbitals (<pyr^HOMO^|SO_2_^LUMO^>, Table 6) clearly correlate with the above-discussed ∆*E*_oi_ terms [57]. In particular, while ∆*E*_oi_ decreases for (n)pyridazine systems and increases for (n)pyrimidine and (n)pyrazine with increasing n, the same trend is given by the corresponding overlaps. In addition, to complete the comparison between the close values between the pyrimidine and pyrazine systems, the slightly more favorable ∆*E*_oi_ for the former is also supported by its larger overlap values and, at the same time, larger charge transfer from the HOMO to the LUMO (Figure 7). It is also important to notice that in all cases, pyrimidine gives a larger <pyr^HOMO^ | SO_2_^LUMO^> than either pyridazine or pyrazine. So, this bent geometry that these circular molecular assemblies adopt with pyrimidine gives rise to the best donor–acceptor interaction between the ring and the SO_2_ molecule. 

Computed noncovalent interaction plots (NCI) help to further understand the stronger interaction in the case of pyridazine and pyrimidine than pyrazine when interacting with SO_2_. In particular, noncovalent interactions can be revealed from the electron density as they are highly nonlocal and manifest in real space as low-gradient isosurfaces with low densities. For such, we use the sign of the second Hessian eigenvalue to determine the kind of interaction, and its strength can be derived from the density on the noncovalent interaction surface [58,59]. In our case, it can be observed that in addition to the N···S interaction for the three pyrazines, in the case of both pyridazine and pyrimidine, there is a weak interaction between one oxygen of SO_2_ and one H of the diazine (Figure 8). This interaction is not present in the case of pyrazine because of its perpendicular geometry.

Overall, the total bonding energies of the supramolecular assemblies decrease in the order of pyridazine, pyrimidine, and pyrazine. Although the N···S bond lengths partially explain these trends, a quantitative energy decomposition analysis (EDA) was necessary to understand the strength of these chalcogen interactions fully. The trends are primarily determined by the attractive electrostatic and orbital interactions, with the strongest interactions observed in pyridazine and weaker interactions in both pyrimidine and pyrazine as the size of the complex increases. In the case of pyrimidine, the most favorable electrostatic interaction is attributed to its most negatively charged nitrogen. Additionally, the orbital interaction, driven by the donor–acceptor interaction between the HOMO of the diazine ring and the LUMO of the SO_2_ molecule, is also most favorable for pyrimidine because of the stronger overlap.

### 2.2. Interaction between Diazines and SO_3_

In this second section, we substitute SO_2_ with SO_3_ and analyze how the interaction with diazines changes. We do not have a symmetric interaction of SO_3_ with the two diazine rings, but it gets closer to one of them, thus approaching the formation of a covalent N-S bond. If we focus on the shorter S···N bond length, with the increase in n, it becomes shorter in the case of pyridazine (Figure 9), whereas it becomes longer in the case of both pyrimidine (Figure 10) and pyrazine (Figure 11), which is a completely opposite trend compared with SO_2_. The only exception is 3pyrimidine/SO_3_ because of its more constrained conformation. The circular systems with pyridazine adopt a planar conformation with D_nh_ symmetry, whereas those with either pyrimidine or pyrazine adopt an almost perpendicular conformation of the rings with C_n_ symmetry. Importantly, for these systems, the sulfur of SO_3_ only directly interacts with one N atom of the ring, as on the other side, we do not have an S···N interaction any longer, and instead, the oxygen atoms are closer to the diazine ring, especially for the more bent pyrimidine and diazine systems.

With respect to the bonding interaction between the diazines and SO_3_, we observe an important change in SO_2_ because now, in all cases, pyrimidine interacts the strongest, with the only exception of 3pyrimidine/SO_3_ mentioned above (Table 7). The other important change is the strain energy, which, in this case, is really large because of the deformation of SO_3_, which loses its planarity and becomes pyramidalized when it interacts with diazine rings by forming the N-S bond. Nonetheless, its magnitude is not determinant in the interaction energies of these systems, which will be further discussed below.

These interaction energies were again analyzed by means of a Kohn–Sham molecular orbital analysis together with a quantitative EDA analysis (Table 8). The pyrimidine systems present the largest ∆*E*_int_ despite Pauli repulsion also being the largest because of the shorter S···N bond lengths compared with either circular systems with pyridazines or pyrazines. However, such large steric repulsion is compensated for by more attractive electrostatic and orbital interactions and even dispersion ones. Importantly, these systems with SO_3_ involve a more covalent interaction than the above ones, with much closer values of electrostatic (47–49%) and orbital interactions (42–47%).

However, in this case, as we do not have a symmetrical system with SO_3_ equally interacting with the diazine on one side as the one on the other, we do not only have one diazine···SO_3_ interaction but two. Therefore, we also analyzed both interactions individually (Table 9 and Table 10). The shorter S···N bond implies a much stronger interaction because a covalent N-S bond is formed. For instance, in the case of 3pyridazine/SO_3_, ∆*E*_int_ amounts to −27.2 and −6.9 kcal mol^−1^ for the shorter and longer S···N interactions. The former has a stronger covalent character compared with the latter, which has a predominant electrostatic character. Importantly, there is a good correlation between ∆*E*_int_ and the S···N distances discussed above. For instance, for n = 6, the S···N goes from 2.084 to 2.002 to 2.037 Å from pyridazine to pyrimidine to pyrazine, whereas ∆*E*_int_ goes from −28.8 to −30.6 to −29.2 kcal mol^−1^, respectively. Such trends are determined by both attractive ∆*V*_elstat_ and ∆*E*_oi_ terms, where those for (n)pyrimidine are the most favorable, despite the repulsive ∆*E*_Pauli_ also being the largest. In contrast, in the case of the longer S···N bond (Table 10), for all diazines, the interaction energy decreases with larger n, thus correlating with the opposite behavior given by the shorter bond. Nonetheless, the fact that their strengths are almost one order of magnitude smaller, their contribution to the overall interaction of the supramolecular cluster is less relevant. 

With respect to the electrostatic interaction, in the case of pyrimidine, the charge transfer from the ring to the SO_3_ is larger because of its more favorable interaction, as mentioned above. Once again, pyrimidine is the diazine that shows the most negatively charged N atoms (Table 11), thus giving rise to a more attractive electrostatic interaction with the even more positively charged S atoms of SO_3_ (+0.625 e). The MEP isosurfaces (Appendix A) show the charge transfer from the diazines to the SO_3_ molecules, with an important decrease in the negative charge on the N atoms, with pyrimidine showing the largest. 

Next, we move to the ∆*E*_oi_ term, which is also responsible for the more attractive interaction. The most determinant orbital interaction is between the HOMO (pyr^HOMO^) of the diazine and the LUMO of SO_3_ (SO_3_^LUMO^, Figure 12). In contrast to SO_2_, the latter involves sigma S-O bonds. Thus, we observe a donor–acceptor interaction between the diazine ring and the SO_3_ molecule. As mentioned above, the interaction orbital increases with larger n, and, in general, this is also the trend supported by the charge transfer from the diazines to the SO_3_ as well as the corresponding overlap between pyr^HOMO^ and SO_3_^LUMO^ (Table 12). However, in this case, we must keep in mind that although this discussion is based on the shorter S···N interaction, that of the longer one also plays a role, thus affecting the trends. Importantly, in line with the more attractive electrostatic interaction, pyrimidine also gives rise to the most attractive orbital interactions.

Finally, and for completeness, NCI plots were also computed for the diazines interacting with SO_3_ (Figure 13), which confirmed the covalent character of the shorter N-S bond formed between the diazines and the SO_3_. This N-S interaction is complemented by weaker H···O noncovalent interactions for the three diazines. On the other hand, we also confirmed that we do not have a longer S···N interaction, but for the three diazines, their oxygen atoms interact with the diazine rings, i.e., a lateral interaction. 

For completeness, nucleophilic and electrophilic Fukui functions were computed, together with the dual descriptor [60,61,62]. This latter is defined as the difference between the Fukui functions for nucleophilic and electrophilic attacks. Thus, the dual descriptor gives a combination of both Fukui functions, positive for locations where a nucleophilic attack is more probable than an electrophilic one, and negative where the electrophilic attack is more probable. Figure 14 shows the dual descriptor plots for the diazines, further supporting the conclusions given above that were obtained from the EDA analysis. It can also be observed how the SO_2_/SO_3_ can perfectly interact with the diazine through the sulfur atom. This study can be further complemented by the electrophilic and nucleophilic Fukui functions (Appendix A). 

## 3. Computational Details

All calculations were carried out using the Amsterdam Density Functional (ADF) program [63]. All stationary points and energies were calculated at the BLYP level of the generalized gradient approximation (GGA) using the exchange functional developed by Becke (B) and the GGA correlation functional developed by Lee, Yang, and Parr (LYP) [64,65]. The DFT-D3(BJ) method developed by Grimme and coworkers [66,67], which contains the damping function proposed by Becke and Johnson [68], was used to describe non-local dispersion interactions. Scalar relativistic effects were accounted for using the zeroth-order regular approximation (ZORA) [69,70,71]. This level is referred to as BLYP-D3(BJ)/TZ2P and has been proven to accurately describe weak interactions [72,73,74]. A large uncontracted optimized TZ2P Slater-type orbitals (STOs) basis set containing diffuse functions were used. The TZ2P all-electron basis set [69], with no frozen-core approximation, is of triple-ζ quality for all atoms and has been augmented with two sets of polarization functions on each atom. The accuracies of the integration grid (Becke grid) and the fit scheme (Zlm fit) were set to VERYGOOD [75,76].

The total bonding energy ∆*E* of the circular supramolecular systems (n)diazine/SO_x_ is defined as [Equation (1)]: ∆*E* = *E*_(n)diazine/SOx_ − n *E*_diazine_ − n *E*_SOx_(1)
where *E*_(n)diazine/SO2_ is the energy of the optimized circular system and *E*_diazine_ and *E*_SO2_ are the energies of the optimized diazine and SO_x_. ∆*E* can be divided into two components by means of the activation strain model (ASM) [77,78,79] [Equation (2)]:∆*E* = ∆*E*_strain_ + ∆*E*_int_(2)
where the strain energy ∆*E*_strain_ is the amount of energy required to deform the individual monomers from their equilibrium structure to the geometry that they acquire in the circular system. The interaction energy ∆*E*_int_ corresponds to the actual energy change when the prepared monomers are combined to form the whole system. 

∆*E*_int_ can be further analyzed through a quantitative energy decomposition analysis (EDA) [80,81], which decomposes ∆*E*_int_ into the classical electrostatic interaction (∆*V*_elstat_) among the unperturbed charge distributions of the deformed monomers, the Pauli repulsion among occupied orbitals (∆*E*_Pauli_), the stabilizing orbital interactions term (∆*E*_oi_) that accounts for charge transfer (i.e., donor–acceptor interactions between occupied orbitals on one moiety and unoccupied orbitals on the other, including the HOMO–LUMO interactions) and polarization (i.e., empty–occupied orbital mixing on one fragment due to the presence of another fragment), and the dispersion correction ∆*E*_disp_ [Equation (3)]:∆*E*_int_ = ∆*V*_elstat_ + ∆*E*_Pauli_ + ∆*E*_oi_ + ∆*E*_disp_(3)

The orbital interaction energy can be further decomposed into the contributions from each irreducible representation Γ of the interacting system [82]. In our planar model systems, the *C*_S_ symmetry partitioning allows us to distinguish between σ and π interactions [Equation (4)]:∆*E*_oi_ = ∆*E*_σ_ + ∆*E*_π_(4)

The electron density distribution is analyzed using the Voronoi deformation density (VDD) method [83] for computing atomic charges. The VDD atomic charge on atom *A* (*Q*_A_^VDD^) is computed as the (numerical) integral of the deformation density in the volume of the Voronoi cell of atom *A* [Equation (5)]. The Voronoi cell of atom *A* is defined as the compartment of space bounded by the bond midplanes on and perpendicular to all bond axes between nucleus *A* and its neighboring nuclei.
(5)$$Q_{A}^{VDD}=-\int_{Voronoi\;cell\;of\;A}^{\;}\left[\rho \left(r\right)-\sum_{B}\rho_{B}\left(r\right)\right]dr$$
where *ρ*(***r***) is the electron density of the molecule and ∑_B_
*ρ*_B_(***r***) is the superposition of atomic densities *ρ*_B_ of a fictitious promolecule without chemical interactions that is associated with the situation in which all atoms are neutral. The interpretation of the VDD charge, *Q*_A_^VDD^ is rather straightforward and transparent: instead of measuring the amount of charge associated with a particular atom A, *Q*_A_^VDD^ directly monitors how much charge flows, because of chemical interactions, out of (*Q*_A_^VDD^ > 0) or into (*Q*_A_^VDD^ < 0) the Voronoi cell of atom A.

## 4. Conclusions

Overall, in our study, we strategically employed organic molecules, specifically diazines, interacting with sulfur oxides via chalcogen bonding giving rise to supramolecular circular structures. This innovative approach is aimed at allowing a better comprehension of the capability of diazine molecules to detect SO_2_/SO_3_ compounds within the atmosphere. With their two nitrogen atoms, diazines exhibit the unique potential to interact with two SO_2_/SO_3_ molecules simultaneously, making them an ideal choice for our circular molecular assembly. The dispersion-corrected DFT Kohn–Sham molecular analysis together with a quantitative energy decomposition analysis highlights the significant importance of these noncovalent interactions. We believe that this newfound knowledge can play a pivotal role in advancing the development of novel materials designed to capture polluting gases, leveraging the unique properties of our molecules.

In particular, we characterized in detail the nature of the S···N interactions in this series of diazines interacting with SO_2_/SO_3_ molecules. For this purpose, we expanded our systems to chains consisting of six to twelve monomers to explore the impact of these larger molecular structures on chalcogen interactions. Both σ-hole and π-hole interactions are further supported by the computed molecular electrostatic potential isosurfaces depending on the interaction between either the lone pair of nitrogen or the ring of the diazines. Noticeably, the trends in the interaction energies of these supramolecular systems when going from pyridazine to pyrimidine to pyrazine are partially determined by the S···N bond lengths and further supported by the attractive electrostatic and orbital interactions. As a whole, the findings of this paper contribute significantly to a better understanding of chalcogen bonding interactions.

## Figures and Tables

**Figure 1 ijms-25-07497-f001:**
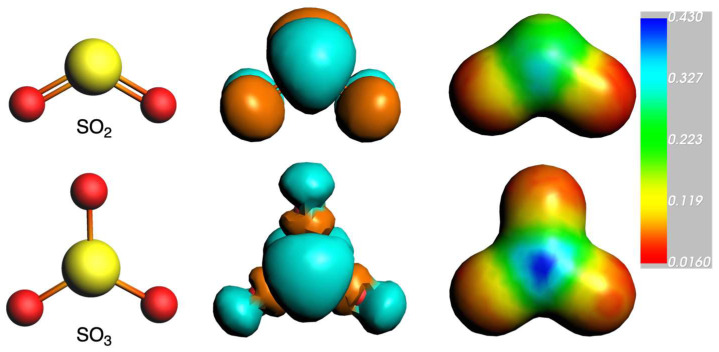
Geometries of SO_2_ and SO_3_ (**left**), together with their LUMO (**center**) and their molecular electrostatic potential isosurfaces (a.u., electronic density isovalue = 0.03 a.u., (**right**)).

**Figure 2 ijms-25-07497-f002:**
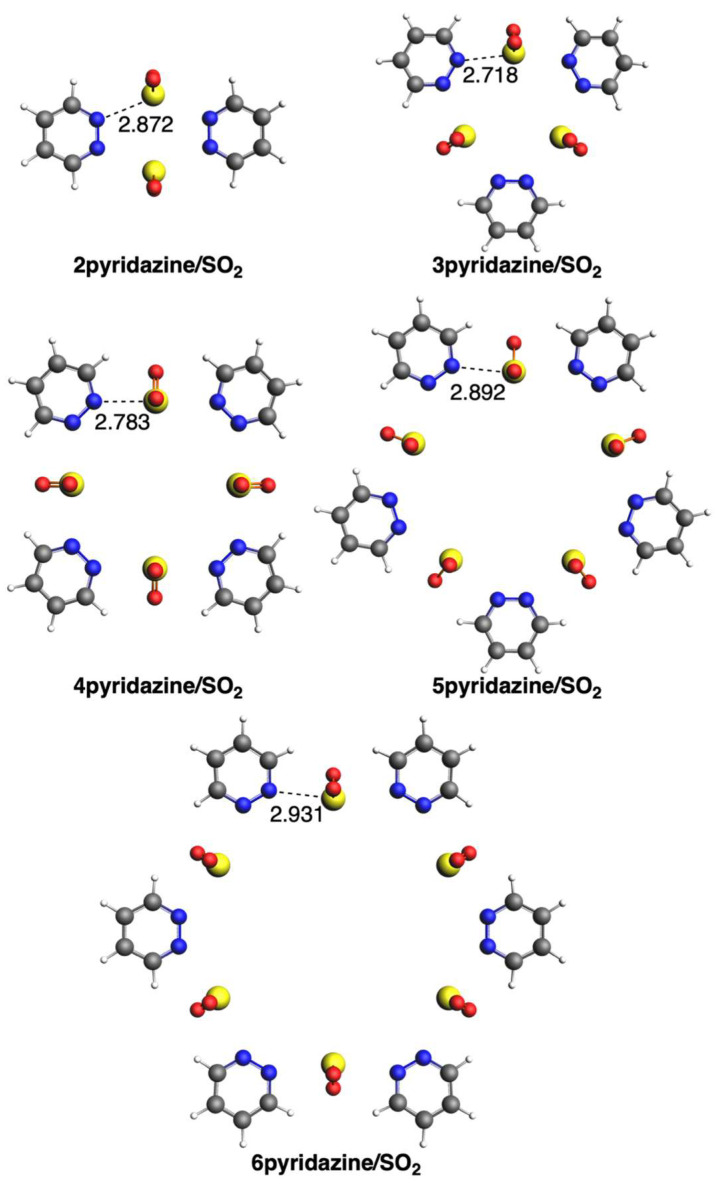
Geometries of complexes with SO_2_ with pyridazine. Bond lengths in Å. Atom colors for N: blue, C: grey, S: yellow, O: red, H: white.

**Figure 3 ijms-25-07497-f003:**
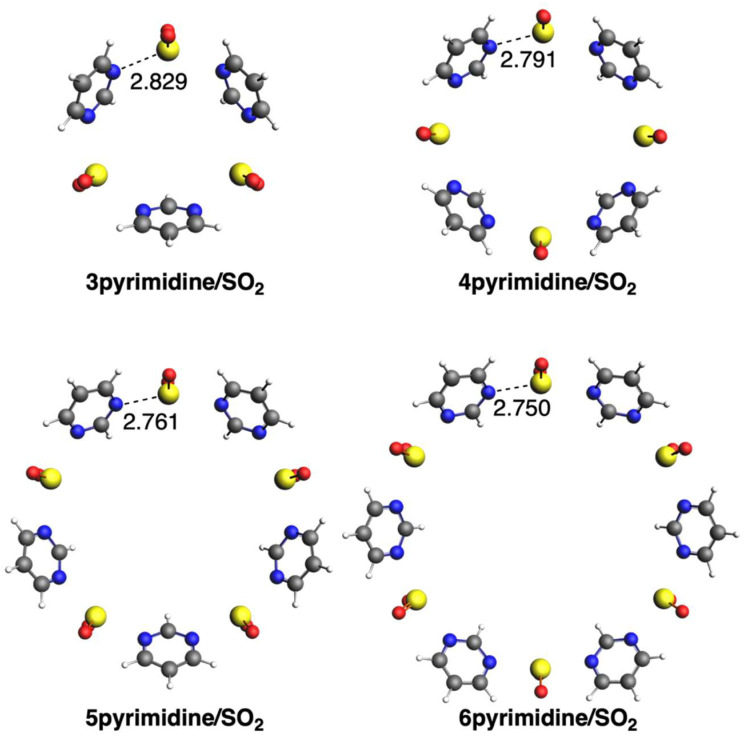
Geometries of complexes with SO_2_ with pyrimidine. Bond lengths in Å. Atom colors for N: blue, C: grey, S: yellow, O: red, H: white.

**Figure 4 ijms-25-07497-f004:**
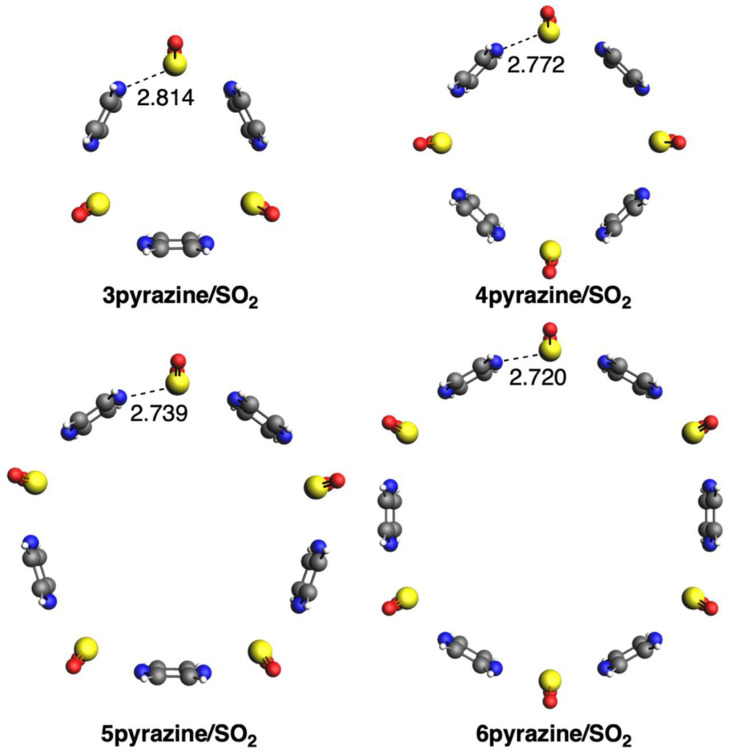
Geometries of complexes with SO_2_ with pyrazine. Bond lengths in Å. Atom colors for N: blue, C: grey, S: yellow, O: red, H: white.

**Figure 5 ijms-25-07497-f005:**
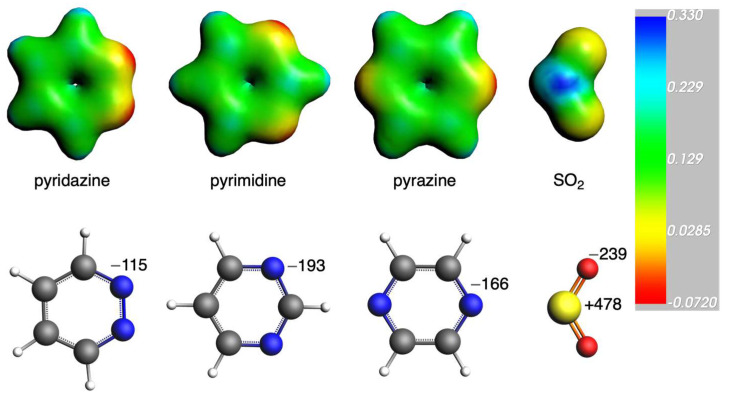
Molecular electrostatic potential isosurface (a.u., electronic density isovalue = 0.03 a.u.) of pyridazine, pyrimidine, pyrazine, and SO_2_ (**top**) and Voronoi deformation density charges (in milli-e, (**bottom**)).

**Figure 6 ijms-25-07497-f006:**
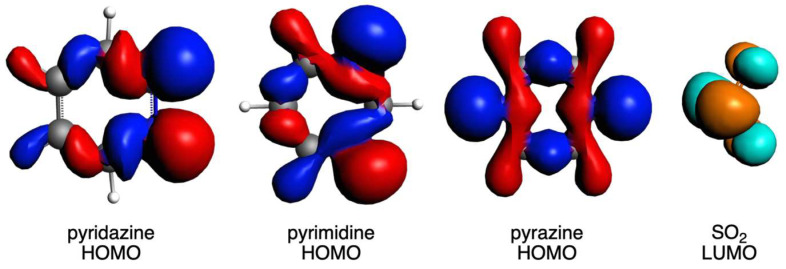
HOMO of pyridazine, pyrimidine, and pyrazine and LUMO of SO_2_.

**Figure 7 ijms-25-07497-f007:**
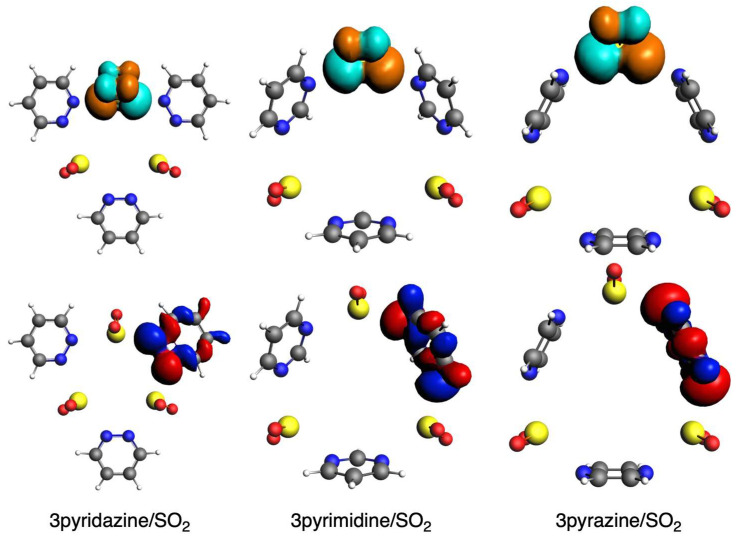
HOMO of ring (**bottom**) and LUMO of SO_2_ (**top**) fragments in complexes with pyridazine, pyrimidine, and pyrazine with 3 units. Red and blue isosurfaces represent positive and negative phases for HOMO, whereas orange and turquoise represent these phases for LUMO.

**Figure 8 ijms-25-07497-f008:**
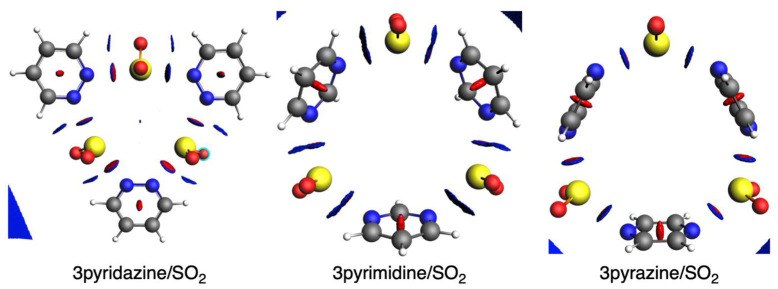
NCI plots (isovalue = 0.5) of SO_2_ complexes with pyridazine, pyrimidine, and pyrazine with 3 units. Color-coded noncovalent interaction (NCI) surfaces (attractive decreasing from blue to green; repulsive increasing from yellow to red).

**Figure 9 ijms-25-07497-f009:**
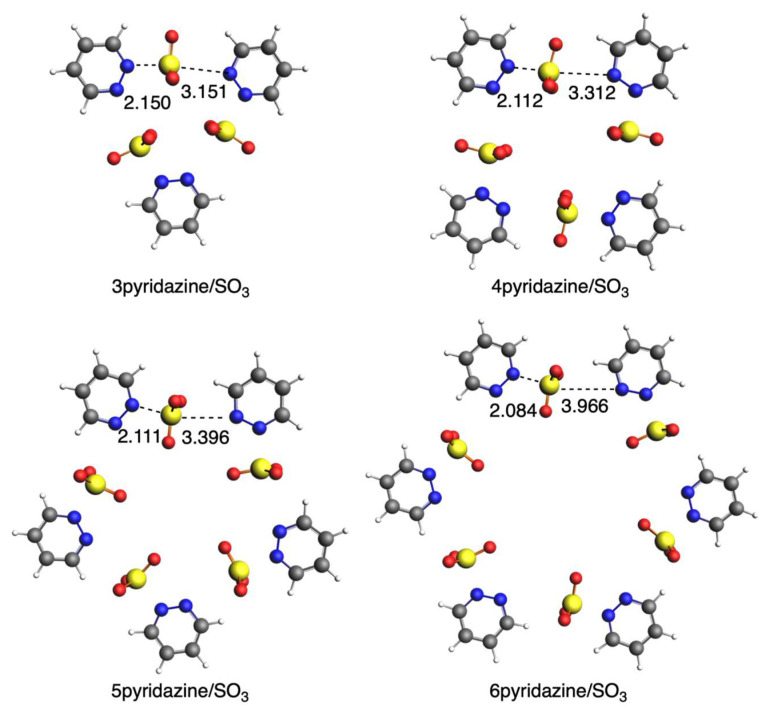
Geometries of complexes with SO_3_ with pyridazine. Bond lengths in Å. Atom colors for N: blue, C: grey, S: yellow, O: red, H: white.

**Figure 10 ijms-25-07497-f010:**
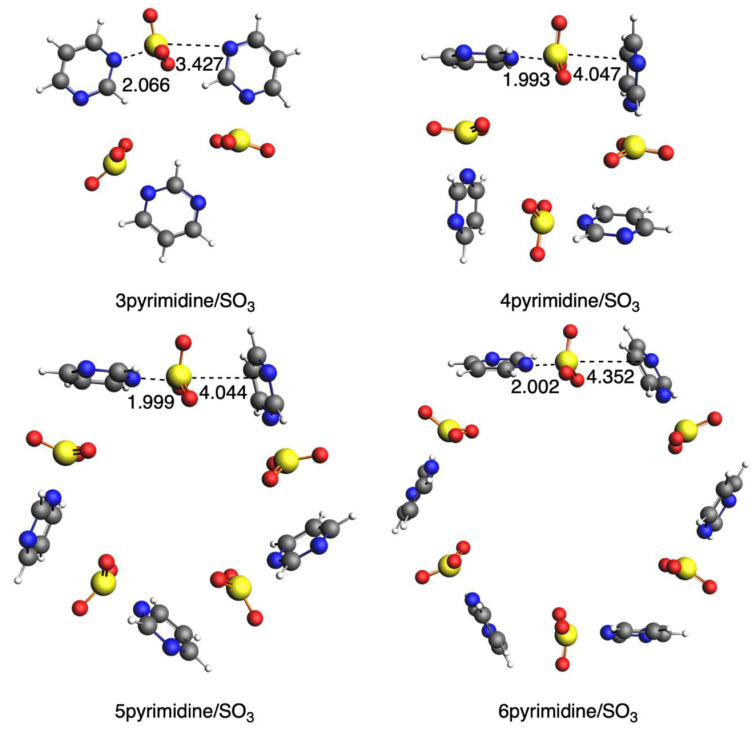
Geometries of complexes with SO_3_ with pyrimidine. Bond lengths in Å. Atom colors for N: blue, C: grey, S: yellow, O: red, H: white.

**Figure 11 ijms-25-07497-f011:**
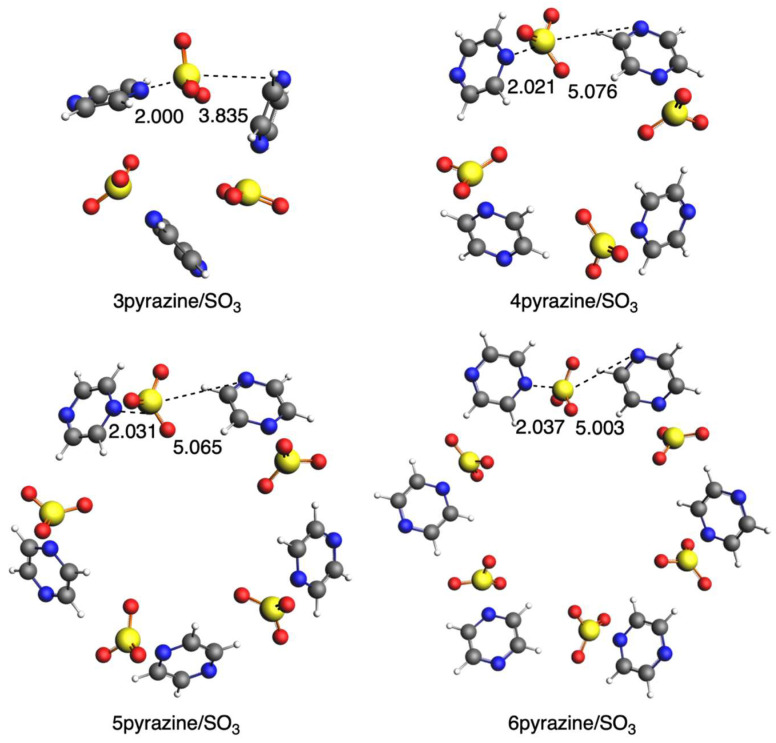
Geometries of complexes with SO_3_ with pyrazine. Bond lengths in Å. Atom colors for N: blue, C: grey, S: yellow, O: red, H: white.

**Figure 12 ijms-25-07497-f012:**
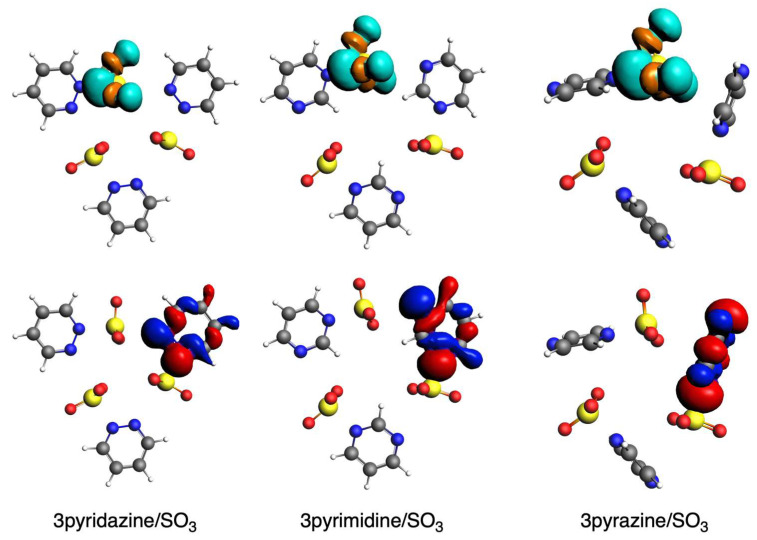
HOMO of ring (**bottom**) and LUMO of SO_3_ (**top**) fragments in complexes with pyridazine, pyrimidine, and pyrazine with 3 units. Red and blue isosurfaces represent positive and negative phases for HOMO, whereas orange and turquoise represent these phases for LUMO.

**Figure 13 ijms-25-07497-f013:**
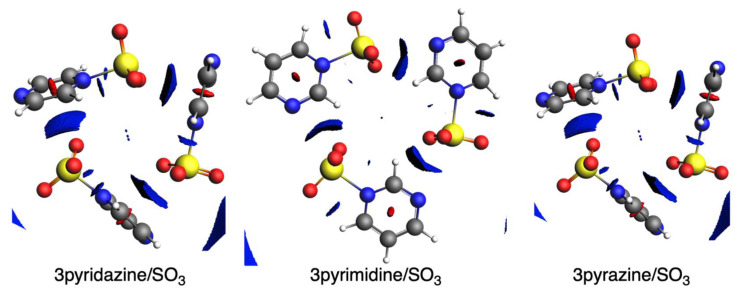
NCI plots (isovalue = 0.5) of SO_3_ complexes with pyridazine, pyrimidine, and pyrazine with 3 units. Color-coded noncovalent interaction (NCI) surfaces (attractive decreasing from blue to green; repulsive increasing from yellow to red).

**Figure 14 ijms-25-07497-f014:**
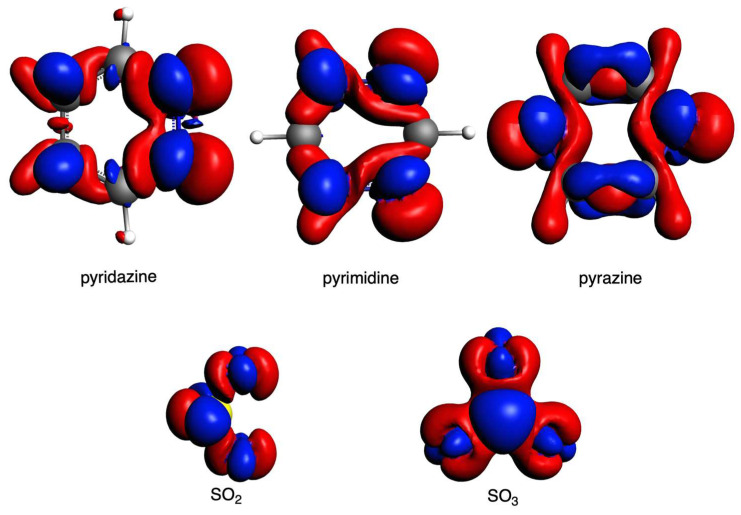
Dual descriptor Fukui functions of the diazines and SO_2_/SO_3_ systems (isovalue = 0.003). Computed at BLYP-D3(BJ)/TZ2P level of theory.

**Table 1 ijms-25-07497-t001:** Total bonding (∆*E*), interaction (∆*E*_int_), and strain (∆*E*_strain_) energies (in kcal mol^−1^) of the (n)pyridazine/SO_2_ complexes with pyridazine, pyrimidine, and pyrazine ^a^.

	Pyridazine			Pyrimidine			Pyrazine		
n	∆*E*	∆*E*_int_	∆*E*_strain_	∆*E*	∆*E*_int_	∆*E*_strain_	∆*E*	∆*E*_int_	∆*E*_strain_
2	−24.7	−25.6	0.9						
3	−46.1	−48.9	2.8	−37.4	−39.0	1.5	−33.4	−34.2	0.8
4	−61.8	−64.7	3.0	−54.7	−56.9	2.2	−48.7	−50.1	1.4
5	−72.0	−74.6	2.5	−71.5	−74.7	3.2	−63.8	−65.7	1.9
6	−80.3	−83.0	2.7	−88.0	−91.9	3.9	−78.6	−80.9	2.3

^a^ Computed at BLYP-D3(BJ)/TZ2P level of theory. ∆*E* = ∆*E*_int_ + ∆*E*_strain_.

**Table 2 ijms-25-07497-t002:** Total bonding energies (∆*E*, in kcal mol^−1^) together with its value divided by the number of molecules for each system (2n) for the (n)pyridazine/SO_2_ complexes with pyridazine, pyrimidine and pyrazine ^a^.

	Pyridazine		Pyrimidine		Pyrazine	
n	∆*E*	∆*E*/2n	∆*E*	∆*E*/2n	∆*E*	∆*E/*2n
3	−46.1	−7.8	−37.4	−6.2	−33.4	−5.6
4	−61.8	−7.7	−54.7	−6.8	−48.7	−6.1
5	−72.0	−7.2	−71.5	−7.1	−63.8	−6.4
6	−80.3	−6.7	−88.0	−7.3	−78.6	−6.6

^a^ Computed at BLYP-D3(BJ)/TZ2P level of theory. ∆*E* = ∆*E*_int_ + ∆*E*_strain_.

**Table 3 ijms-25-07497-t003:** Energy decomposition analysis (in kcal mol^−1^) of the (n)pyridazine/SO_2_ complexes with pyridazine, pyrimidine, and pyrazine ^a^.

n	∆*E*_int_	∆*E*_Pauli_	∆*V*_elstat_	∆*E_oi_*	∆*E*_disp_
pyridazine					
3	−48.9	116.4	−85.1	−55.5	−24.7
4	−64.7	142.8	−105.4	−69.5	−32.6
5	−74.6	143.8	−109.4	−69.8	−39.2
6	−83.0	152.7	−116.9	−70.8	−48.0
pyrimidine					
3	−39.0	86.3	−62.4	−39.7	−23.2
4	−56.9	130.9	−95.3	−61.4	−31.1
5	−74.7	178.3	−130.1	−83.6	−39.3
6	−91.9	221.6	−161.9	−104.1	−47.5
pyrazine					
3	−34.2	85.8	−58.4	−38.6	−23.1
4	−50.1	127.2	−89.7	−57.2	−30.3
5	−65.7	178.6	−125.8	−79.7	−38.8
6	−80.9	229.0	−161.1	−101.6	−47.3

^a^ Computed at BLYP-D3(BJ)/TZ2P level of theory. ∆*E*_int_ = ∆*E*_Pauli_ + ∆*V*_elstat_ + ∆*E*_oi_ +∆*E*_disp_.

**Table 4 ijms-25-07497-t004:** Energy decomposition analysis (in kcal mol^−1^) of the interaction between one molecule of pyridazine, pyrimidine, and pyrazine and SO_2_ at the same geometry as the circular systems ^a^.

n	∆*E*_int_	∆*E*_Pauli_	∆*V*_elstat_	∆*E_oi_*	∆*E*_disp_
pyridazine					
3	−10.4	19.2	−15.1	−10.8	−3.7
4	−10.1	17.8	−14.0	−10.0	−3.8
5	−8.9	14.3	−11.6	−7.8	−3.7
6	−8.1	12.6	−10.4	−6.5	−3.8
pyrimidine					
3	−7.7	14.3	−10.7	−7.6	−3.7
4	−8.6	16.3	−12.3	−8.9	−3.8
5	−9.0	17.8	−13.4	−9.6	−3.8
6	−9.3	18.5	−13.9	−10.0	−3.9
pyrazine					
3	−6.7	13.5	−9.5	−7.1	−3.6
4	−7.6	15.8	−11.5	−8.3	−3.7
5	−8.1	17.9	−12.9	−9.3	−3.8
6	−8.3	19.0	−13.8	−9.7	−3.8

^a^ Computed at BLYP-D3(BJ)/TZ2P level of theory. ∆*E*_int_ = ∆*E*_Pauli_ + ∆*V*_elstat_ + ∆*E*_oi_ +∆*E*_disp_.

**Table 5 ijms-25-07497-t005:** VDD charges (in milli-e) of complexes of SO_2_ with pyridazine, pyrimidine, and pyrazine.

	pyridazine			
	3	4	5	6
N	−58	−67	−80	−84
S	433	442	457	459
O	−304	−311	−295	−284
O	−330	−285	−277	−278
	pyrimidine			
	3	4	5	6
N	−157	−150	−145	−143
S	434	431	428	426
O	−296	−303	−309	−309
O	−282	−288	−294	−296
	pyrazine			
	3	4	5	6
N	−134	−122	−119	−114
S	429	423	420	419
O	−287	−297	−303	−306
O	−287	−297	−303	−306

**Table 6 ijms-25-07497-t006:** Energies of fragment molecular orbitals (in eV), their overlap, and their Gross Mulliken populations (in au) of complexes of SO_2_ with pyridazine, pyrimidine, and pyrazine.

	pyridazine			
	3	4	5	6
E(pyr^HOMO^)	−5.20	−5.23	−5.27	−5.29
E(SO_2_^LUMO^)	−4.61	−4.60	−4.59	−4.58
overlap	0.078	0.074	0.064	0.058
pop(Hpyr^HOMO^)	1.89	1.89	1.91	1.91
pop(SO_2_^LUMO^)	0.17	0.16	0.13	0.11
	pyrimidine			
	3	4	5	6
E(pyr^HOMO^)	−5.82	−5.80	−5.79	−5.79
E(SO_2_^LUMO^)	−4.57	−4.58	−4.59	−4.59
overlap	0.079	0.085	0.088	0.091
pop(Hpyr^HOMO^)	1.92	1.91	1.91	1.90
pop(SO_2_^LUMO^)	0.13	0.15	0.16	0.17
	pyrazine			
	3	4	5	6
E(pyr^HOMO^)	−5.75	−5.74	−5.72	−5.72
E(SO_2_^LUMO^)	−4.54	−4.55	−4.56	−4.56
overlap	0.062	0.070	0.073	0.075
pop(Hpyr^HOMO^)	1.94	1.93	1.92	1.92
pop(SO_2_^LUMO^)	0.13	0.14	0.15	0.16

**Table 7 ijms-25-07497-t007:** Total bonding (∆*E*), interaction (∆*E*_int_), and strain (∆*E*_strain_) energies (in kcal mol^−1^) of the (n)pyridazine/SO_3_ complexes with pyridazine, pyrimidine, and pyrazine ^a^.

	Pyridazine			Pyrimidine			Pyrazine		
n	∆*E*	∆*E*_int_	∆*E*_strain_	∆*E*	∆*E*_int_	∆*E*_strain_	∆*E*	∆*E*_int_	∆*E*_strain_
3	−69.4	−87.9	18.5	−76.8	−100.7	23.9	−86.7	−116.7	30.0
4	−98.2	−127.7	29.4	−117.7	−161.8	44.0	−111.2	−148.3	37.0
5	−118.6	−155.8	37.1	−145.1	−199.4	54.3	−137.5	−182.7	45.2
6	−143.5	−194.6	51.1	−171.7	−236.3	64.6	−163.2	−217.1	53.9

^a^ Computed at BLYP-D3(BJ)/TZ2P level of theory. ∆*E* = ∆*E*_int_ + ∆*E*_strain_.

**Table 8 ijms-25-07497-t008:** Energy decomposition analysis (in kcal mol^−1^) of the (n)pyridazine/SO_3_ complexes with pyridazine, pyrimidine, and pyrazine ^a^.

n	∆*E*_int_	∆*E*_Pauli_	∆*V*_elstat_	∆*E_oi_*	∆*E*_disp_
pyridazine					
3	−87.9	307.4	−195.1	−166.6	−33.7
4	−127.7	457.4	−285.0	−259.4	−40.7
5	−155.8	555.7	−342.4	−318.2	−50.8
6	−194.6	716.5	−432.6	−423.2	−55.2
pyrimidine					
3	−100.7	396.8	−244.8	−220.6	−32.1
4	−161.8	656.6	−384.4	−377.7	−56.3
5	−199.4	805.0	−472.3	−465.4	−66.6
6	−236.3	950.2	−560.5	−553.4	−72.6
pyrazine					
3	−116.7	478.7	−281.3	−273.9	−40.2
4	−148.3	600.9	−358.2	−349.6	−41.4
5	−182.7	726.8	−435.1	−422.9	−51.5
6	−217.1	857.3	−513.9	−497.5	−63.1

^a^ Computed at BLYP-D3(BJ)/TZ2P level of theory. ∆*E*_int_ = ∆*E*_Pauli_ + ∆*V*_elstat_ + ∆*E*_oi_ +∆*E*_disp_.

**Table 9 ijms-25-07497-t009:** Energy decomposition analysis (in kcal mol^−1^) of the interaction between one molecule of diazine and SO_3_ (shorter S···N distance) at the same geometry as the circular systems ^a^.

n	∆*E*_int_	∆*E*_Pauli_	∆*V*_elstat_	∆*E_oi_*	∆*E*_disp_
pyridazine					
3	−27.2	95.0	−60.9	−55.6	−5.8
4	−28.5	107.5	−67.6	−62.4	−5.9
5	−27.8	105.8	−66.0	−61.7	−5.9
6	−28.8	115.6	−71.0	−67.3	−6.0
pyrimidine					
3	−28.6	126.2	−77.6	−70.8	−6.5
4	−30.7	156.1	−92.8	−87.4	−6.7
5	−30.6	153.5	−91.3	−86.1	−6.7
6	−30.6	152.2	−90.8	−85.4	−6.7
pyrazine					
3	−29.9	152.0	−90.3	−84.9	−6.7
4	−29.4	143.0	−85.8	−80.0	−6.7
5	−29.2	138.9	−83.7	−77.8	−6.6
6	−29.2	136.8	−82.7	−76.8	−6.6

^a^ Computed at BLYP-D3(BJ)/TZ2P level of theory. ∆*E*_int_ = ∆*E*_Pauli_ + ∆*V*_elstat_ + ∆*E*_oi_ +∆*E*_disp_.

**Table 10 ijms-25-07497-t010:** Energy decomposition analysis (in kcal mol^−1^) of the interaction between one molecule of diazine and SO_3_ (longer S···N distance) at the same geometry as the circular systems ^a^.

n	∆*E*_int_	∆*E*_Pauli_	∆*V*_elstat_	∆*E_oi_*	∆*E*_disp_
pyridazine					
3	−6.9	5.5	−5.9	−3.2	−3.3
4	−5.9	6.2	−5.6	−3.3	−3.3
5	−5.0	5.0	−4.1	−2.5	−3.4
6	−3.3	3.8	−2.2	−2.0	−2.9
pyrimidine					
3	−4.4	5.6	−4.4	−2.1	−3.6
4	−4.2	7.9	−3.2	−2.5	−6.4
5	−4.4	7.3	−3.1	−2.6	−6.0
6	−3.9	6.0	−2.5	−2.4	−5.0
pyrazine					
3	−2.8	7.4	−3.2	−1.5	−5.6
4	−2.6	7.2	−3.2	−3.6	−3.1
5	−2.4	6.4	−2.8	−2.9	−3.1
6	−2.2	5.9	−2.4	−2.4	−3.3

^a^ Computed at BLYP-D3(BJ)/TZ2P level of theory. ∆*E*_int_ = ∆*E*_Pauli_ + ∆*V*_elstat_ + ∆*E*_oi_ +∆*E*_disp_.

**Table 11 ijms-25-07497-t011:** VDD charges (in milli-au) of complexes of SO_3_ with pyridazine, pyrimidine, and pyrazine.

pyridazine				
	3	4	5	6
N	28	33	32	39
S	541	543	539	533
N	−39	−40	−42	−45
pyrimidine				
	3	4	5	6
N	−34	−22	−22	−22
S	541	527	527	527
N	−142	−159	−157	−154
pyrazine				
	3	4	5	6
N	0	−5	−6	−6
S	532	537	538	537
N	−135	−137	−138	−138

**Table 12 ijms-25-07497-t012:** Energies of fragment molecular orbitals (in eV), their overlap, and their Gross Mulliken populations (in au) of complexes of SO_3_ with pyridazine, pyrimidine, and pyrazine.

pyridazine				
	3	4	5	6
E(pyr^HOMO^)	−5.12	−5.11	−5.10	−5.09
E(SO_3_^LUMO^)	−5.22	−5.34	−5.37	−5.49
Overlap	0.189	0.195	0.196	0.202
pop(Hpyr^HOMO^)	1.76	1.73	1.72	1.69
pop(SO_3_^LUMO^)	0.35	0.40	0.40	0.43
pyrimidine				
	3	4	5	6
E(pyr^HOMO^)	−5.67	−5.67	−5.67	−5.67
E(SO_3_^LUMO^)	−5.48	−5.77	−5.76	−5.75
Overlap	0.204	0.213	0.212	0.211
pop(Hpyr^HOMO^)	1.74	1.70	1.71	1.71
pop(SO_3_^LUMO^)	0.44	0.51	0.51	0.50
pyrazine				
	3	4	5	6
E(pyr^HOMO^)	−5.65	−5.65	−5.65	−5.65
E(SO_3_^LUMO^)	−5.69	−5.62	−5.60	−5.59
Overlap	0.200	0.197	0.195	0.195
pop(Hpyr^HOMO^)	1.72	1.73	1.74	1.74
pop(SO_3_^LUMO^)	0.51	0.49	0.48	0.47

## Data Availability

All data is available at the Appendix A.

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
