# Peer review of "Chalcogen Noncovalent Interactions between Diazines and Sulfur Oxides in Supramolecular Circular Chains"

_ijms, 2024, doi:10.3390/ijms25137497_

Round 1
Reviewer 1 Report
Comments and Suggestions for Authors
Dear Authors,
After thoroughly examining your manuscript, I would like to express my positive feedback regarding your submission. Your study is of potential interest to the readers of the International Journal of Molecular Sciences. However, I believe there is room for improvement, and I would like to request some revisions based on the following remarks:
· The applied level of theory should be elaborated. If possible, please provide references where the applied level of theory had been used.
· Please introduce the names of all columns of Tables 2 and 3. What is the meaning of bold numbers?
· Scheme 1 should be named Figure 1. This Figure should also contain a color bar with corresponding maximal and minimal MEP values.
· Values of MEP in Figures 4, 5 and 12 should be provided in different color. Currently, it is tough to read them.
· Since the MEP descriptor has been extensively applied, please introduce a paragraph about it and mention a few papers of Prof. Politzer and coworkers. They made a tremendous contribution to the development and popularization of that descriptor.
· Please explain the NCI plots presented in Figure 14. How have these figures been obtained? They remind me of reduced density gradient (RDG) surfaces. What are the meanings of different colors...?
Upon addressing all of the aforementioned comments, I would be pleased to review your manuscript again.
Best regards
Author Response
REVIEWER 1
After thoroughly examining your manuscript, I would like to express my positive feedback regarding your submission. Your study is of potential interest to the readers of the International Journal of Molecular Sciences. However, I believe there is room for improvement, and I would like to request some revisions based on the following remarks:
ANSWER: We thank the Reviewer for the positive evaluation of our manuscript.
- The applied level of theory should be elaborated. If possible, please provide references where the applied level of theory had been used.
ANSWER: The level of theory used, i.e., ZORA-BLYP-D3(BJ)/TZ2P, has largely proven to correctly address weak interactions, like the noncovalent interactions addressed in our work.
Several benchmark studies have been done in the past, like P. Vermeeren, L. P. Wolters, G. Paragi, C. Fonseca Guerra, ChemPlusChem 2021, 86. 812; T. A. Hamlin, J. Poater, C. Fonseca Guerra, F. M. Bickelhaupt, Phys. Chem. Chem. Phys.2017, 19, 16969, which showed that our used level of theory ZORA-BLYP-D3(BJ)/TZ2P is well suited for accurately describing noncovalent interactions. This conclusion is even reinforced by the benchmark study of Head-Gordon et al. (Mol. Phys. 2017, 115, 2315) entitled “Thirty years of density functional theory in computational chemistry: an overview and extensive assessment of 200 density functionals”. These references have been introduced in the manuscript to further support our choice. In addition, this level of theory with a GGA functional has also proven to correctly address HOMO-LUMO gap energies, as shown in Phys. Chem. Chem. Phys. 2013, 15, 16408, or J. Chem. Theory Comput. 2014, 10, 4432.
- Please introduce the names of all columns of Tables 2 and 3. What is the meaning of bold numbers?
ANSWER: As suggested by the Reviewer, we have clarified the names of all tables enclosed in the manuscript. In particular, we have improved the captions, and clarified that the bold numbers refer to the value of n in (n)pyridazine/SO2or (n)pyridazine/SO3 molecular names.
- Scheme 1 should be named Figure 1. This Figure should also contain a color bar with corresponding maximal and minimal MEP values.
ANSWER: The figure has been modified accordingly and named as Figure 1. The rest of the figures have been renumbered.
- Values of MEP in Figures 4, 5 and 12 should be provided in different color. Currently, it is tough to read them.
ANSWER: We have increased the size of the legend in order to easily read the values.
- Since the MEP descriptor has been extensively applied, please introduce a paragraph about it and mention a few papers of Prof. Politzer and coworkers. They made a tremendous contribution to the development and popularization of that descriptor.
ANSWER: We apologize for having omitted the relevant works by Politzer and coworkers with MEP on both sigma- and pi-holes. A new sentence has been introduced in the Introduction together with 4 new references relevant to the work.
- Please explain the NCI plots presented in Figure 14. How have these figures been obtained? They remind me of reduced density gradient (RDG) surfaces. What are the meanings of different colors...?
ANSWER: The noncovalent interaction (NCI) plots have been further explained and added the colors used in the captions of the corresponding figures. NCI is also based on the electron density and makes use of the second Hessian eigenvalue.
Upon addressing all of the aforementioned comments, I would be pleased to review your manuscript again.
We hope that with the above responses, our manuscript is now acceptable for publication in IJMS.
Sincerely yours,
Prof. Dr. Jordi Poater
Universitat de Barcelona
jordi.poater@ub.edu
Reviewer 2 Report
Comments and Suggestions for Authors
The paper by Poater and coworkers on chalcogen bonding between diazines and sulfur-oxides is written in excellent English language and presents very interesting work based on several computational methodologies. In my opinion, the authors presented a very good strategy for studying noncovalent interactions in supramolecular systems with larger number of small molecules. I believe that the authors wanted to present this strategy on several interesting examples that they have studied, and these examples also have additional value because they are relevant for environmental chemistry. My only major objection is that despite the very elaborate analysis of noncovalent interaction in their systems, the authors had not made a summary of their results on these particular systems and gave a summary regarding the trends that were noticed. The best place to do this is, in my opinion, at the end of sections 2.1. and 2.2., after all the analyses are presented. This is particularly the case for the observed trends for SO2 interactions going from n = 3 to n = 6; for pyridazine these trends are different than those for pyrimidine and pyrazine. The explanations for this (and other trends) need to be clearly singled out and emphasized. Moreover, the conclusion and perhaps the abstract should also contain at least some details regarding the pyridazine, pyrimidine and pyrazine interactions. My recommendation is therefore that this excellent paper should be accepted after this minor revision.
There are a few minor points:
- Please add colored scale for electrostatic potential surfaces in Scheme 1.
- Molecules appear to be smaller in some figures, and bigger in others. I believe that they should be of equal size in all figures.
- Why were the electrostatic potentials plotted on such large isovalue of electronic density? They are usually plotted at isovalue of 0.001 a.u., and here the authors used 0.03 a.u.
- Why are the MEP isosurfaces of already formed complexes presented in the Figure 5? It is typical to use electrostatic potentials only of individual molecules.
- It should be clearly stated in the abstract and conclusion that SO3 interactions are significantly more covalent than SO2 interactions.
- Perhaps the energy decomposition for longer S∙∙∙N bonds should also be presented in the main text instead of Supporting Information (I believe that this is Table S2). The best place for this is, in my opinion, right after the EDA for shorter S∙∙∙N bonds (Table 9).
Author Response
REVIEWER 2
The paper by Poater and coworkers on chalcogen bonding between diazines and sulfur-oxides is written in excellent English language and presents very interesting work based on several computational methodologies. In my opinion, the authors presented a very good strategy for studying noncovalent interactions in supramolecular systems with larger number of small molecules. I believe that the authors wanted to present this strategy on several interesting examples that they have studied, and these examples also have additional value because they are relevant for environmental chemistry. My only major objection is that despite the very elaborate analysis of noncovalent interaction in their systems, the authors had not made a summary of their results on these particular systems and gave a summary regarding the trends that were noticed. The best place to do this is, in my opinion, at the end of sections 2.1. and 2.2., after all the analyses are presented. This is particularly the case for the observed trends for SO2interactions going from n = 3 to n = 6; for pyridazine these trends are different than those for pyrimidine and pyrazine. The explanations for this (and other trends) need to be clearly singled out and emphasized. Moreover, the conclusion and perhaps the abstract should also contain at least some details regarding the pyridazine, pyrimidine and pyrazine interactions. My recommendation is therefore that this excellent paper should be accepted after this minor revision.
ANSWER: We thank the Reviewer to consider that our work is interesting. As suggested by the Reviewer, we have summarized the results at the end of the discussion of the supramolecular assemblies between diazines and SO2. A whole new paragraph has been added focusing on the trends in interaction energies going from pyridazine to pyrimidine and to pyrazine. We refer that the computed S···N bond lengths partially determine the trends, but that the EDA is required in order to fully understand what happens. In particular, both attractive electrostatic and orbital interactions play a determinant role: pyrimidine shows the most negatively-charged N atom, thus giving rise to the most attractive electrostatic interaction. In addition, its stronger donor-acceptor interaction between the HOMO of the ring and the LUMO of SO2 also gives the more attractive orbital interaction term. These observations regarding the interactions have also been further emphasized in both the abstract and conclusions.
There are a few minor points:
- Please add colored scale for electrostatic potential surfaces in Scheme 1.
ANSWER: Scheme 1 has been corrected accordingly with the introduction of a legend for MEP isosurfaces. And it has been renamed as new Figure 1.
- Molecules appear to be smaller in some figures, and bigger in others. I believe that they should be of equal size in all figures.
ANSWER: Sizes of figures were changes when converted into the template, thus we apologize for it. We have resized all figures in order to try to make them of equal size.
- Why were the electrostatic potentials plotted on such large isovalue of electronic density? They are usually plotted at isovalue of 0.001 a.u., and here the authors used 0.03 a.u.
ANSWER: All MEP isosurfaces have been plotted by means of Amsterdam Density Functional software, that is our main computational software not only to visualize the results, but also to perform the quantum chemical computations at DFT level. By default, the isovalue for MEP is 0.03 a.u. to correctly visualize the color scale, with a smaller isovalue the figure does not allow a correct visualization of the different zones.
- Why are the MEP isosurfaces of already formed complexes presented in the Figure 5? It is typical to use electrostatic potentials only of individual molecules.
ANSWER: As suggested by the Reviewer, Figure 5 and Figure 12 have been removed from the main manuscript and moved to the supporting information file. The discussion is now focused on the MEP isosurfaces of the individual molecules.
- It should be clearly stated in the abstract and conclusion that SO3 interactions are significantly more covalent than SO2 interactions.
ANSWER: As suggested by the Reviewer, we have stressed the clearly more covalent character when diazines interact with SO3 than with SO2.
- Perhaps the energy decomposition for longer S∙∙∙N bonds should also be presented in the main text instead of Supporting Information (I believe that this is Table S2). The best place for this is, in my opinion, right after the EDA for shorter S∙∙∙N bonds (Table 9).
ANSWER: As suggested by the Reviewer, the EDA analysis corresponding to the longer S···N bond has been enclosed in the main manuscript as new Table 10. And extra discussion has been added too.
We hope that with the above responses, our manuscript is now acceptable for publication in IJMS.
Sincerely yours,
Prof. Dr. Jordi Poater
Universitat de Barcelona
jordi.poater@ub.edu

Reviewer 3 Report
Comments and Suggestions for Authors
In this work, the authors investigated the noncovalent chalcogen interactions between SO2/SO3 and diazines using DFT calculations. It involves constructing supramolecular circular chains of up to 12 molecules to evaluate ability of diazine to detect SO2/SO3 in the atmosphere. The interaction energies are influenced primarily by Pauli steric repulsion in σ-hole/π-hole interactions. These findings help define the essential characteristics of chalcogen bonding for capturing sulfur oxides.
I would like to ask the authors to consider the minor comments below.
1. The electrostatic potential has been used in this work for the analysis, which is not very informative. Can the authors add analysis using Fukui function to show the chemical reactivity?
2. page 16, Figure 12
There are several issues in this figure: (a) the resolution of this figures (and other figures) is too low. (b) the unit of the colorbar is missing. (c) significant figures in the colorbar should be consistent.
3. page 17, Figure 13
HOMO and LUMO orbitals should not be plotted together, which makes the figure hard to read and distinguish. Can the authors also explain why LUMOs are more delocalized over space than HOMOs in this figure?
4. page 17, Table 11
It should be mentioned in the context that HOMO-LUMO gaps evaluated at the DFT levels are typically underestimated.
5. page 18, line 357
“This level is referred to as BLYP-D3(BJ)/TZ2P”
Why BLYP is chosen in this study? Using hybrid or range-separated functionals can lead to improved accuracy.
Comments on the Quality of English LanguageNo major language or grammar problem found.
Author Response
REVIEWER 3
In this work, the authors investigated the noncovalent chalcogen interactions between SO2/SO3 and diazines using DFT calculations. It involves constructing supramolecular circular chains of up to 12 molecules to evaluate ability of diazine to detect SO2/SO3 in the atmosphere. The interaction energies are influenced primarily by Pauli steric repulsion in σ-hole/π-hole interactions. These findings help define the essential characteristics of chalcogen bonding for capturing sulfur oxides.
I would like to ask the authors to consider the minor comments below.
- The electrostatic potential has been used in this work for the analysis, which is not very informative. Can the authors add analysis using Fukui function to show the chemical reactivity?
ANSWER: The bonding analysis between the diazines and SO2/SO3 has been performed by means of a Kohn-Sham molecular orbital analysis together with an energy decomposition analysis. The electrostatic interaction term (∆Velstat) has been justified by means of the computed Voronoi Deformation Density charges (VDD), that clearly show how the negative charge on the N atoms of the diazines determine the trends in ∆Velstat. The discussion has been further complemented by means of the MEP isosurfaces, that clearly support the discussed trends. We agree with the Reviewer that this information is not very informative, at difference to the VDD charges that give quantitative data. However, MEP plots have been largely used in such sigma- and pi-hole analyses. For the same reason, the orbital interaction term is analyzed by means of the Gross Mulliken populations, the fragment molecular orbital energies, and with the overlaps between the interacting orbitals in the donor-acceptor interaction. All this later data is again quantitative and is also very informative. Nonetheless, we have complemented the discussion by adding the Fukui functions, as suggested by the Reviewer with new Figure 14 and Figure S3. The electrophilic/nucleophilic Fukui functions perfectly support the conclusions given by the EDA analysis regarding the interaction between the diazines and the SOx molecules. The Fukui functions nicely complements our conclusions, and for this reason we thank the Reviewer for this advice.
- page 16, Figure 12
There are several issues in this figure: (a) the resolution of this figures (and other figures) is too low. (b) the unit of the colorbar is missing. (c) significant figures in the colorbar should be consistent.
ANSWER: MEP isosurfaces are in atomic units. We have added it in the corresponding figure captions. We have limited the discussion of the MEP to the fragments used, not the whole systems, as these are more related to the electrostatic interaction term discussed in the energy decomposition analysis. The legends of these figures have been increased. But, as stated above, the main discussion on the electrostatic interaction is based on the VDD charges.
- page 17, Figure 13
HOMO and LUMO orbitals should not be plotted together, which makes the figure hard to read and distinguish. Can the authors also explain why LUMOs are more delocalized over space than HOMOs in this figure?
ANSWER: As suggested by the Reviewer, both figures enclosing the HOMO and LUMO fragment molecular orbitals of SO2 and SO3 have been modified, and now top figures include the LUMOs whereas bottom figures the HOMOs. The larger delocalization of the LUMO is directly related to the larger size of the S atom in either SO2 or SO3. The density is delocalized along the two or three S-O bonds. At difference, in case of diazines, HOMOs are more localized on the interacting N atom.
- page 17, Table 11
It should be mentioned in the context that HOMO-LUMO gaps evaluated at the DFT levels are typically underestimated.
ANSWER: Sorry, but with this point we do not agree with the Reviewer. All computations have been performed at the ZORA-BLYP-D3(BJ)/TZ2P level. And Baerends and coworkers proved that this level of theory with a GGA functional correctly addresses HOMO-LUMO gap energies, as shown in Phys. Chem. Chem. Phys. 2013, 15, 16408, or J. Chem. Theory Comput. 2014, 10, 4432.
- page 18, line 357
“This level is referred to as BLYP-D3(BJ)/TZ2P”
Why BLYP is chosen in this study? Using hybrid or range-separated functionals can lead to improved accuracy.
ANSWER: The level of theory used, i.e., ZORA-BLYP-D3(BJ)/TZ2P, has largely proven to correctly address weak interactions, like the noncovalent interactions addressed in our work.
Several benchmark studies have been done in the past, like P. Vermeeren, L. P. Wolters, G. Paragi, C. Fonseca Guerra, ChemPlusChem 2021, 86. 812; T. A. Hamlin, J. Poater, C. Fonseca Guerra, F. M. Bickelhaupt, Phys. Chem. Chem. Phys.2017, 19, 16969, which showed that our used level of theory ZORA-BLYP-D3(BJ)/TZ2P is well suited for accurately describing noncovalent interactions. This conclusion is even reinforced by the benchmark study of Head-Gordon et al. (Mol. Phys. 2017, 115, 2315) entitled “Thirty years of density functional theory in computational chemistry: an overview and extensive assessment of 200 density functionals”. These references have been introduced in the manuscript to further support our choice.
We hope that with the above responses, our manuscript is now acceptable for publication in IJMS.
Sincerely yours,
Prof. Dr. Jordi Poater
Universitat de Barcelona
jordi.poater@ub.edu

Reviewer 4 Report
Comments and Suggestions for Authors
In this submission to IJMS, the authors investigate noncovalent chalcogen interaction between SO2/SO3 and diazines using dispersion-corrected DFT Kohn-Sham molecular orbital approaches with quantitative energy decomposition analyses. The authors constructed supramolecular circular chains of up to 12 molecules to check the capability of diazine molecules to detect SO2/SO3 compounds within the atmosphere. The authors find that interaction energies with an increasing number of molecules are mainly determined by the Pauli steric repulsion involved in these σ-hole/π-hole interactions. The authors find that π-hole interactions are supported by the charge transfer from the diazines to the SO2/SO3 molecules. The authors conclude that their results establish the fundamental characteristics of chalcogen bonding based on its strength and nature, of relevance for the capture of sulfur oxides.
I find this manuscript to be of interest to computational chemists as well as readers of this journal. As such, I am generally supportive of publication with a few required edits. Specifically, there has been prior work on σ-hole/π-hole interactions with dispersion-corrected functionals for the related halogen family, which should be noted: J. Chem. Theory Comput. 2013, 9, 1918−1931 and J. Chem. Theory Comput. 2018, 14, 180–190. In particular, these prior studies also examined σ-hole/π-hole interactions with dispersion-corrected functionals to understand the mechanisms responsible for binding in these compounds. With this minor edit, I would be willing to re-review this manuscript for subsequent publication in IJMS.
Author Response
REVIEWER 4
In this submission to IJMS, the authors investigate noncovalent chalcogen interaction between SO2/SO3 and diazines using dispersion-corrected DFT Kohn-Sham molecular orbital approaches with quantitative energy decomposition analyses. The authors constructed supramolecular circular chains of up to 12 molecules to check the capability of diazine molecules to detect SO2/SO3 compounds within the atmosphere. The authors find that interaction energies with an increasing number of molecules are mainly determined by the Pauli steric repulsion involved in these σ-hole/π-hole interactions. The authors find that π-hole interactions are supported by the charge transfer from the diazines to the SO2/SO3 molecules. The authors conclude that their results establish the fundamental characteristics of chalcogen bonding based on its strength and nature, of relevance for the capture of sulfur oxides.
I find this manuscript to be of interest to computational chemists as well as readers of this journal. As such, I am generally supportive of publication with a few required edits. Specifically, there has been prior work on σ-hole/π-hole interactions with dispersion-corrected functionals for the related halogen family, which should be noted: J. Chem. Theory Comput. 2013, 9, 1918−1931 and J. Chem. Theory Comput. 2018, 14, 180–190. In particular, these prior studies also examined σ-hole/π-hole interactions with dispersion-corrected functionals to understand the mechanisms responsible for binding in these compounds. With this minor edit, I would be willing to re-review this manuscript for subsequent publication in IJMS.
ANSWER: We thank the Reviewer for the positive evaluation of our manuscript. The two suggested references on σ-hole/π-hole interactions have been added into the manuscript, in particular when referring to the halogen bonding in the Introduction.
We hope that with the above responses, our manuscript is now acceptable for publication in IJMS.
Sincerely yours,
Prof. Dr. Jordi Poater
Universitat de Barcelona
jordi.poater@ub.edu

Round 2
Reviewer 3 Report
Comments and Suggestions for Authors
The authors have correctly addressed all my technical remarks. Further review is not needed.
Comments on the Quality of English LanguageNo major language or grammar problem found.
Reviewer 4 Report
Comments and Suggestions for Authors
This paper can be accepted in its current form.